# RethinkMCTS: Refining Erroneous Thoughts in Monte Carlo Tree Search for Code Generation

## Abstract

LLM agents enhanced by tree search algorithms have shown significant performance in code generation. However, existing search methods generally operate directly in the code language space, leading to suboptimal search quality due to ignoring the reasoning process behind the code. Specifically, two key challenges remain largely unaddressed: 1) A lack of exploration for the reasoning process, which is essential for high-reasoning-demand tasks like code generation, and 2) Inadequate search quality due to the absence of refinement mechanism. In this paper, we introduce RethinkMCTS, a framework that explores and refines the reasoning process for generating code. Specifically, we employ MCTS to search for the thoughts before code generation and integrate MCTS with a refinement mechanism called *rethink*, which incorporates fine-grained code execution feedback to refine erroneous thoughts during the search. It ensures the search path aligns with better reasoning, improving overall search quality. Through extensive experiments, we demonstrate that RethinkMCTS outperforms previous search-enhanced and feedback-enhanced code generation baselines. On the HumanEval dataset, it boosts the pass@1 of GPT-3.5-turbo from **70.12** to **89.02** and that of GPT-4o-mini from **87.20** to **94.51**. By conducting thought-level exploration and integrating the *rethink* mechanism, it significantly enhances the search quality of the entire search tree[1].

## 1 Introduction

Coding has become an increasingly valuable skill in the digital information era (Liu et al., 2023). As the capabilities of large language models (LLMs) continue to impress, research has increasingly focused on enhancing their code generation abilities (Luo et al., 2023; Zheng et al., 2023; Gong et al., 2024). Early efforts concentrate on pre-training or fine-tuning language models specifically on vast amounts of code data (Li et al., 2023; Roziere et al., 2023). With the growing power of general LLMs and the need for external tools and resources such as compilers and code libraries in code generation (Zhou et al., 2024; Kannan et al., 2023), utilizing general LLMs as agents to improve code generation through algorithmic design has

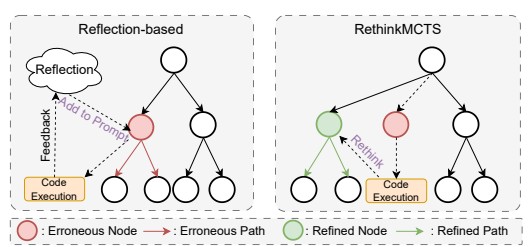

Figure 1: Comparison between reflection-based methods and RethinkMCTS. Reflection-based methods would maintain the error in the path, while RethinkMCTS would refine erroneous thoughts and continue along a better path.

emerged as a promising direction (Jain et al., 2023; Ugare et al., 2024).

In research where LLMs are used as agents for code generation, search methods have been widely applied and have demonstrated remarkable effectiveness (Zhou et al., 2023a; DeLorenzo et al., 2024). These methods often explore various possibilities through search techniques. Despite achieving notable results, directly exploring and refining at the code language space—be it at the token

---

[1]Resources are available at https://anonymous.4open.science/r/RethinkMCTS-D748/.

level (Zhang et al., 2023), line level (Kulal et al., 2019) or even the entire program level (Zhou et al., 2023a)—may limit the effectiveness of the search by overlooking the underlying reasoning process behind the code.

Specifically, existing search approaches for code generation face two key limitations: 1) The lack of exploration for the reasoning process for code generation as a high-reasoning-demand task. Studies, such as chain-of-thought (Wei et al., 2022) and tree of thoughts (Yao et al., 2024), show that explicitly modeling the reasoning process leads to better results. Research by Tang et al. (2023) further highlighted that LLMs are better equipped for semantic reasoning than symbolic reasoning. However, for code generation, a high-reasoning-demand task (Cook et al., 2018), current work has yet to explore the thoughts (reasoning) behind the generated code. 2) Inadequate search quality due to the lack of refinement mechanism. A refinement mechanism holds great potential for search algorithms in code generation. From the code environment perspective, the detailed feedback obtained from code execution is highly informative and plays a crucial role in refining erroneous outputs (Wang et al., 2022; Zhong et al., 2024). Previous work has also shown that LLMs can refine their previous outputs when provided with external feedback (Zhou et al., 2023b; Gou et al., 2023), making a refinement mechanism highly suitable for this task. From the tree search perspective, since the search tree is typically built incrementally, refining earlier outputs ensures that the exploration remains on more optimal paths, thereby improving the overall search quality. However, such a refinement mechanism has yet to be successfully integrated into code generation search algorithms. Some methods tried self-reflection to summarize past errors (Zhou et al., 2023a; Shinn et al., 2024), however, as shown in Figure 1, they would remain erroneous actions in the exploration path, leading subsequent searches to continue along incorrect trajectories.

In this work, we address these limitations by focusing on the thought process of code generation. We propose RETHINKMCTS, a framework that explicitly searches reasoning steps before generating code and refines erroneous thoughts based on code execution feedback. Specifically, RETHINKMCTS begins by employing the MCTS algorithm to explore reasoning paths before generating code and then generates the code based on these reasoning thoughts. After executing the code, we perform block-level analysis on the failed public test cases and acquire the verbal execution feedback. Following this, we introduce a refinement mechanism called *rethink*, which makes the LLM refine erroneous thoughts based on the feedback. As shown in Figure 1, this enables the search algorithm to continue exploring along corrected paths, ultimately enhancing the search tree's quality. To further guide action evaluation in the MCTS search process, we propose a dual evaluation approach to ensure effective code selection, particularly when public test cases alone are insufficient. Overall, RETHINKMCTS connects three stages of code generation—reasoning (before), coding (during), and refining (after)—through its *rethink* mechanism. Our main contributions can be summarized as follows:

- **Reasoning-to-Code Search Framework for Code Generation**: Our framework employs a multi-step thinking process combined with single-step code generation using Monte Carlo Tree Search (MCTS) to explore various strategies for code generation. A combination of verbal and scalar feedback guides the MCTS tree generation. To the best of our knowledge, we are the first to search and refine the thought process behind code to enhance LLMs on code generation.

- **Refining Erroneous Thoughts in MCTS**: We introduce the *rethink* mechanism into MCTS to refine erroneous thoughts using detailed verbal feedback from code execution, allowing the search to follow higher-quality traces. Different from reflection-based methods that summarize past mistakes without changing current erroneous reasoning, our approach directly refines flawed thoughts, ensuring the search proceeds along more optimal trajectories.

- **Introducing Detailed Feedback and Dual Evaluation for Refinement**: Block-level analysis is introduced as the detailed feedback of code execution, guiding the refinement of faulty thought. Additionally, a dual evaluation method—using both public test cases and LLM self-evaluations—is used to ensure effective code selection, particularly when public test cases alone cannot fully assess the code's correctness.

## 2 PRELIMINARIES

### 2.1 PROBLEM FORMULATION

The task is code generation, and we follow the setup from previous work by Zhang et al. (2023). Specifically, for an LLM-based agent, the input consists of a problem statement $P$ and a set of public test cases $T_{\text{pub}}$. The goal is to develop a code generation agent model $M$ that produces the correct code $C \sim M(P, T_{\text{pub}})$ to solve the given problem. Each test case of $T_{\text{pub}}$ is defined by an input-output pair. To evaluate the effectiveness of the generated code, we also retain a set of private test cases $T_{\text{priv}}$, which remains hidden from the agent during the code generation process. The primary metric for evaluating the quality of the generated code is whether it can pass these private test cases.

### 2.2 BLOCK-LEVEL CODE ANALYSIS

Executing buggy code in an executor can only provide standard error information. If the code runs without crashing but produces incorrect outputs, there is often little to no error feedback available. However, since code is quite structured (Chevalier et al., 2007), it is possible to extract detailed execution feedback through a more organized analysis. We follow previous work by Zhong et al. (2024) to get a block-level code analysis.

In static code analysis, the code could be divided into basic blocks (Larus, 1999). A basic block is defined as a linear sequence of code containing a single entry point and a single exit point (Flow, 1994; Alfred et al., 2007). We first acquire the control-flow graph (CFG) of the code, and then a public test case is fed into this graph to produce an execution trace of the test, $[B_1, B_2, ..., B_n]$, where each node within the CFG corresponds to a basic block. We execute these blocks one by one and track all variable state changes in the trace. These blocks and variables are collected and then provided to the LLM to perform a block-level analysis, assessing whether each block is correct or faulty. We show an example of the analysis process in the Appendix B.5.

### 2.3 MONTE CARLO TREE SEARCH

Monte Carlo Tree Search (MCTS) is a heuristic search algorithm that achieves great success in decision-making tasks (Silver et al., 2016). It combines the exploration of tree search with the randomness of Monte Carlo simulations to make decisions in complex environments. It initials the problem description as the root node and moves down the tree by selecting actions (child nodes) until the leaf node according to the Upper Confidence Bound (UCB) (Silver et al., 2017) algorithm that balances exploration and exploitation. Then, the MCTS would generate new child nodes for the chosen leaf node. For the newly generated node, the MCTS would simulate it until the terminal state and assign an evaluation reward to this node. Finally, the reward is backpropagated along the way back to the root node. Each node would update its value based on the newly collected reward.

## 3 RETHINKMCTS

**Overview**    The motivation of RETHINKMCTS is to search and refine the thought process during code generation based on the feedback from the coding environment, ultimately guiding the agent toward the correct solution. To achieve this, we take an LLM as the agent to generate the thoughts and the code, as well as refine the thoughts based on the code execution feedback. We employ Monte Carlo Tree Search (MCTS) as the search algorithm to balance exploration and exploitation during the search for thoughts. More importantly, we introduce a *rethink* mechanism, utilizing detailed feedback from code execution to refine erroneous thoughts. This allows the search to follow improved paths and, in turn, enhances the quality of the search. The framework is shown in Figure 2, and we provide the pseudocode in Algorithm 1 in the Appendix C. Our design has the following key features:

- **Tree Search for Thought Process**: We employ tree search to explore the thought process of writing code. After multiple reasoning steps, code is generated based on the accumulated thoughts.

- **Block-Level Analysis Feedback**: We use block-level analysis of the code as the fine-grained feedback from code execution.

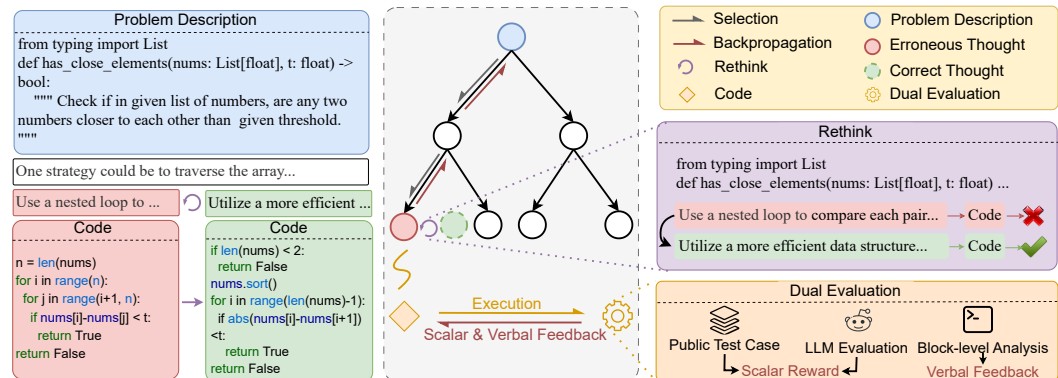

Figure 2: Overview of RETHINKMCTS. We use MCTS to explore different thoughts before generating code. We obtain block-level analysis as verbal feedback through a code executor and use the verbal feedback from failed test cases to refine the thoughts, thereby improving the overall quality of the search tree.

- **Rethink Mechanism**: We introduce a *rethink* mechanism that leverages feedback from the code execution to refine and improve the quality of the reasoning process.

- **Dual Evaluation**: In our evaluation phase, we propose a dual evaluation approach, wherein both public test cases and LLM evaluation are used to assess the generated code, ultimately helping to identify high-quality solutions.

These key features are integrated into operations in RETHINKMCTS, *selection, expansion, evaluation, verbal feedback, backpropagation, and rethink*.

**Selection** In MCTS, the selection step balances exploration and exploitation by iteratively choosing the actions that are most promising for further expansion. This process continues until a leaf node is reached. Each node is selected based on a score derived from the number of visits $N(s)$ and the stored value of the state-action pair $Q(s, a)$, where the state $s$ is the problem description and prior thoughts, and action $a$ represents the new thought associated with the node. Every node's retained value $Q(s, a)$ is the maximum reward obtained by starting in $s$ and taking action $a$. For scoring, we employ P-UCB (Silver et al., 2017), an enhanced version of the UCB algorithm, to compute the overall score for each node:

$$\text{P-UCB}(s, a) = Q(s, a) + \beta(s) \cdot p(a \mid s) \cdot \frac{\sqrt{\log(N(s))}}{1 + N(s')}, \tag{1}$$

where $s'$ is the state reached by taking action $a$ in $s$; $N(s)$ is the visited times of the node; $p(a \mid s)$ is the probability that thought $a$ is the next thought given the problem description and previous thoughts $s$, which is proposed by the LLM agent. $\beta$ is the weight for exploration, which depends on the number of visit of $s$, defined as

$$\beta(s) = \log\left(\frac{N(s) + c_{\text{base}} + 1}{c_{\text{base}}}\right) + c, \tag{2}$$

where $c_{\text{base}}$ is a hyperparameter; $c$ is the exploration weight.

At each state or node, the selection process chooses the action with the highest P-UCB value, and repeats this process until a leaf node is reached.

**Expansion** After selecting a leaf node, the expansion step generates its child nodes to explore different possible actions. We define the search action space as potential thoughts or strategies for writing the code. To make use of the feedback obtained from code execution, we handle the expansion in two scenarios:

- If the current leaf node evaluation has failed public test cases, the expansion step incorporates the verbal feedback $f$ from these failed test cases into the prompt. The LLM agent then proposes

multiple subsequent thoughts $z$ and assigns each thought a reasonableness score $e$, as represented by $p(a|s)$ in Eq. (1). The output is based on prior thoughts and the current verbal feedback, i.e., $[(z^1, e^1), \ldots, (z^k, e^k)] \sim p((z, e)^{(1 \cdots k)}|s, f)$.

- If the current leaf node evaluation passes all public test cases, the expansion step directs the agent to propose subsequent thoughts without additional feedback, i.e., $[(z^1, e^1), \ldots, (z^k, e^k)] \sim p((z, e)^{(1 \cdots k)}|s)$.

**Evaluation**  The primary goal of the evaluation is to estimate the likelihood that the current node will successfully complete the final task. Some previous works refer to the evaluation step in MCTS as "simulation" (Zhou et al., 2023a; Hao et al., 2023), as it typically involves simulating the progression from the node's intermediate state to a terminal state and evaluating the terminal state. For the task of code generation, we search for the thoughts and evaluate with the code generated following the thoughts, meaning we generate complete code based on the currently produced thoughts and use the evaluation of the code as the reward.

In code generation, a natural approach is to use the pass rate of public test cases (Zhang et al., 2023) as the reward. However, the limitation of this method is that public test cases cover only a part of the test set. When multiple code outputs pass all the public test cases, some may still fail to fully solve the problem, making it difficult to differentiate between them. Some earlier efforts have tried to address this by generating additional test cases to cover a broader range of scenarios (Huang et al., 2023b; Zhou et al., 2023a), but this method is costly and does not guarantee that the generated test cases are accurate. To overcome this challenge, we propose a dual evaluation approach. Once all public test cases are passed, we further instruct the LLM to provide a self-assessed comprehensive score, $v^{\text{llm}}$, to evaluate the code's correctness in solving the whole problem.

$$\text{reward} = \begin{cases} v^{\text{test}}, & \text{if } 0 \leq v^{\text{test}} < 1 \\ a \times v^{\text{test}} + b \times v^{\text{llm}}, & \text{if } v^{\text{test}} = 1 \end{cases}, \tag{3}$$

where $v^{\text{test}}$ is the pass rate on public test cases; $v^{\text{llm}}$ is the LLM's self-evaluation score. $a$ and $b$ controls the weight of two parts.

The reward in this context is a scalar value, used to calculate the Q-value at each node and to determine the score during the selection phase. However, in code generation, the compiler and executor can return detailed error messages, and various code analysis tools can provide more granular insights into the code. These details about the code are crucial for making modifications but can not be captured in a scalar reward. Therefore, alongside the scalar reward, we also integrate verbal feedback.

**Verbal Feedback**  When the generated code fails to pass a public test case, human programmers typically diagnose the issue by examining details such as variable values during execution. In the context of solving code generation tasks with search algorithms, relying solely on scalar feedback based on the pass rate of public test cases lacks detailed information. Therefore, we incorporate verbal feedback in the MCTS process. Specifically, as described in Sec. 2.2, we perform block-level analysis when the code fails a public test case and store the resulting information as verbal feedback in the current node. This feedback is then utilized in both the *expansion* and *rethink* phases.

**Backpropagation**  In MCTS, backpropagation refers to the process of updating the Q-values of all nodes along the path from the current node to the root node using the rewards obtained from the evaluation. Beyond using scalar feedback to update the values of parent nodes, verbal feedback is also stored in the current leaf node for use in subsequent *expansion* and *rethink* phases.

**Rethink**  When the code fails to pass a public test case, we can obtain block-level analysis as detailed verbal feedback on the execution. How can we leverage such fine-grained feedback to produce correct code? We propose to use this feedback to make the LLM "rethink", meaning to regenerate the current erroneous thought based on the feedback to avoid generating the incorrect code. As shown in Figure 2, the leaf node is re-generated by $z^{\text{new}} \sim p(z|s, f, z^{\text{old}})$. It is important to emphasize that we do not regenerate the parent nodes in the trace for two key reasons: 1) The parent nodes have already accumulated rewards over multiple rounds of evaluation from all their child nodes, and regenerating them would invalidate the previously gathered rewards. 2) The parent

node has already gone through its own *rethink* process. This means that either the parent node did not encounter failing public test cases during its evaluation or has already been refined through the *rethink* process. Therefore, we do not revisit the parent nodes when performing the *rethink* on a leaf node.

The advantage of introducing *rethink* is twofold. From the code generation perspective, *rethink* refines the reasoning process behind writing code, thus would ultimately lead to better code. From the MCTS perspective, it refines the current action or current node. Since the MCTS tree is built incrementally, improving the quality of the current action allows the LLM to explore more optimal paths in the vast search space, thereby enhancing the overall search quality of the tree. Through the *rethink* mechanism, we seamlessly integrate the process of refining the reasoning of code generation with the MCTS search process.

## 4 EXPERIMENT SETTINGS

**Datasets** We evaluate RETHINKMCTS and baseline methods on two widely used benchmark datasets: APPS (Hendrycks et al., 2021) and HumanEval (Chen et al., 2021). The APPS dataset contains three levels of difficulties: introductory, interview, and competition. We evaluate all the methods on the formal 100 problems of each difficulty. Since the APPS dataset does not distinguish between public and private test cases, we split each program's test cases evenly into two sets, following the approach of Zhang et al. (2023). The first set is used as the public test cases for the algorithms to optimize the pass rate, and the second set is used as the private test cases for evaluating the generated programs. We use *pass rate* and *pass@1* as the evaluation metrics for code correctness following Zhang et al. (2023). *Pass rate* is the average percentage of private test cases successfully passed by the generated code across all problems, and *pass@1* measures the percentage of problems where the generated programs pass all private test cases, which is the most widely adopted metric in the literature of code generation (Austin et al., 2021; Chen et al., 2021; Dong et al., 2023).

**Baselines** To illustrate the effectiveness of RETHINKMCTS, we compare two kinds of code generation methods. The first kind is feedback-enhanced, which uses the code execution feedback to refine codes iteratively: LDB (Zhong et al., 2024), Reflexion (Shinn et al., 2024). The second kind is tree search-enhanced methods: PG-TD (Zhang et al., 2023), ToT (Yao et al., 2024), LATS (Zhou et al., 2023a) and RAP (Hao et al., 2023).

**Implementation** We pick GPT-3.5-turbo and GPT-4o-mini as the backbone models to compare different algorithms. For search-enhanced methods, including RETHINKMCTS, we set the maximum number of children of any node to be 3. For MCTS-based methods, we set the hyperparameters in Eq. (2) $c_{base}$ to be 10 and $c$ to be 4 following previous work by Zhang et al. (2023). And we set the $a$ and $b$ in Eq. (3) to be $(0.8, 0.2)$ and we compare performances under different settings in Sec. 5. We set the maximum number of rollouts or simulation times to be 16. For efficiency, following Zhang et al. (2023), we cache all the codes generated during tree search and finally output the one with the highest evaluation score. For LDB, we set the maximum number of debug times to be 10, as in the original paper (Zhong et al., 2024). For ToT, we define the search action as thought and prompt each node to generate a complete code based on the node's thoughts, similar to RETHINKMCTS, but without incorporating detailed feedback or the *rethink* mechanism. Additionally, the original ToT is designed to handle simple problems that can be clearly divided into discrete steps. Here, we adapt it for code generation in a manner similar to RETHINKMCTS.

## 5 RESULTS

**Overall Performance** We present the overall performance in Table 1, where we can see that RE-THINKMCTS outperforms all baseline models across both datasets. Additionally, by comparing them with the original base model, both feedback-enhanced and tree search-enhanced methods show significant performance improvements, demonstrating the effectiveness of exploring different strategies and using detailed feedback from code execution. Notably, the ToT baseline achieves impressive performance, demonstrating the advantage of searching for the reasoning process during the coding process. However, ToT is a general framework that is designed originally for simpler problems like Game of 24 (Yao et al., 2024), which could be clearly divided into discrete steps

| | | Pass Rate (%) | | | Pass@1 (%) | | | |
|---|---|---|---|---|---|---|---|---|
| | | APPS Intro. | APPS Inter. | APPS Comp. | APPS Intro. | APPS Inter. | APPS Comp. | HumanEval |
| **GPT-3.5-turbo** | Base | 50.43 | 40.57 | 23.67 | 29 | 19 | 9 | 70.12 |
| | PG-TD | 60.89 | 50.80 | 26.50 | 40 | 25 | 8 | 76.22 |
| | ToT | 63.21 | 63.49 | 26.30 | 37 | 33 | 11 | 84.15 |
| | LATS | 54.06 | 45.86 | 21.83 | 36 | 20 | 7 | 79.88 |
| | RAP | 43.22 | 43.32 | 22.83 | 21 | 14 | 8 | 71.95 |
| | LDB | 56.68 | 46.78 | 21.00 | 35 | 22 | 8 | 81.09 |
| | Reflexion | 53.20 | 45.58 | 17.50 | 35 | 21 | 7 | 71.95 |
| | RETHINKMCTS | **67.09** | **68.65** | **29.50** | **45** | **38** | **13** | **89.02** |
| **GPT-4o-mini** | Base | 56.56 | 52.40 | 35.00 | 35 | 29 | 16 | 87.20 |
| | PG-TD | 65.87 | 70.37 | 43.16 | 45 | 46 | 27 | 91.46 |
| | ToT | 74.34 | 71.83 | 42.50 | 55 | 47 | 27 | 93.29 |
| | LATS | 69.46 | 67.65 | 35.83 | 50 | 45 | 19 | 93.29 |
| | RAP | 64.24 | 57.25 | 37.67 | 39 | 32 | 20 | 87.20 |
| | LDB | 60.64 | 60.78 | 40.33 | 40 | 38 | 23 | 90.85 |
| | Reflexion | 60.65 | 56.87 | 38.00 | 40 | 31 | 18 | 90.85 |
| | RETHINKMCTS | **76.60** | **74.35** | **42.50** | **59** | **49** | **28** | **94.51** |

Table 1: Performances of RETHINKMCTS and baselines on APPS and HumanEval. RETHINKM-CTS achieves the best performance across all the datasets with the maximum number of rollouts for tree search algorithms being 16.

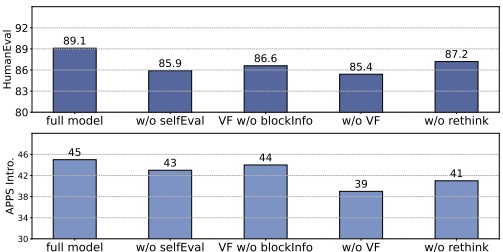

Figure 3: Ablation study of block-level analysis (blockInfo), rethink mechanism, verbal feedback (VF), and self-evaluation with GPT-3.5-turbo as the backbone.

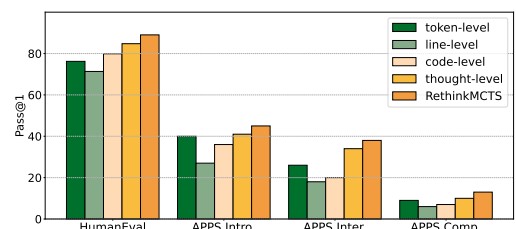

Figure 4: Performance comparison between different search granularity. For advanced models like GPT-3.5-turbo, it's better to explore at the thought level.

and search for each step, essentially different from our idea of searching and refining the reasoning thoughts or strategies of writing code. It does not utilize feedback during the search process. Its search quality is inferior to RETHINKMCTS, underscoring the importance of feedback in refining the thought process.

**Ablation Study** We conduct ablation studies to remove each of our model's components and reevaluate them. For the verbal feedback component, we compare two variants: the first variant removes verbal feedback entirely, relying only on the scalar reward (w/o VF), while the second replaces verbal feedback with standard error messages or incorrect code outputs, omitting block-level analysis information (VF w/o blockInfo). The results using GPT-3.5-turbo as the backbone model are shown in Figure 3, and the results on GPT-4o-mini are presented in Appendix A. The chart shows that each module contributes to the model's overall performance. Verbal feedback has the most significant impact, as the *rethink* mechanism we proposed is primarily based on feedback from code execution. Without this feedback, providing instructions for *rethink* alone would not be sufficient for the model to refine thoughts effectively. This result highlights that detailed feedback from code execution is the key to refining erroneous reasoning in the context of code generation. In fact, previous studies (Huang et al., 2023c) have noted that LLMs lack the ability to self-correct their reasoning without external feedback.

Additionally, we can see that for the HumanEval dataset, block-level analysis information significantly affects performance ($89.1 \rightarrow 86.6$), while for the APPS dataset, the impact is smaller. We hypothesize that this is due to the fewer public test cases in HumanEval compared to APPS (average 2.8 (HumanEval) vs. 27.52 (APPS Introductory) public test cases), making fine-grained analysis of each test case crucial for *rethink* to refine erroneous thought in HumanEval. In contrast, the ample number of public test cases in APPS Introductory allows the model to find the issues with only standard error information. This is also why dual evaluation is crucial for HumanEval, as the limited number of public test cases is insufficient to fully assess the quality of a code snippet. In such cases,

| | Pass Rate (%) | | | Pass@1 (%) | | | |
|---|---|---|---|---|---|---|---|
| | APPS Intro. | APPS Inter. | APPS Comp. | APPS Intro. | APPS Inter. | APPS Comp. | HumanEval |
| Direct Evaluation | 76.60 | 74.34 | 42.50 | 59 | 49 | 28 | 94.51 |
| Self-generated Tests | 77.32 | 75.80 | 47.23 | 59 | 44 | 28 | 93.29 |

Table 2: The performance comparison between using Direct Self-evaluation and Self-generating test evaluation.

it becomes necessary to introduce LLM to reevaluate the code. Finally, the *rethink* mechanism we proposed significantly enhances the results. This improvement stems from that *rethink* enabling the use of fine-grained block-level analysis in verbal feedback, effectively correcting logical errors in the reasoning process.

**Search Granularity Study** RETHINKMCTS conducts a thought-level search during code generation. Here, we compare the action spaces for MCTS, specifically examining different levels of search granularity: token, line, code, and thought. The experimental results with GPT-3.5-turbo as backbone are presented in Figure 4, and the results on GPT-4o-mini are presented in Appendix A.

As shown in the figure, the thought-level search is more effective in finding viable code compared to token, line, and code-level searches. This demonstrates that for advanced LLMs like GPT-3.5-turbo, exploring the reasoning process is beneficial (Zhang et al., 2024b; Huang & Chang, 2022). Additionally, we observe that token-level searching performs better than line and code-level searching. This is due to the fact that with a limited number of search iterations, token-level searches allow fewer constraints on the early tokens, thus uncovering more possibilities compared to line and code-level searches. Finally, although thought-level search yields the best results among different granularities, its effectiveness is further enhanced in RETHINKMCTS by introducing detailed feedback and *rethink* mechanism, making the search over thoughts in the code generation process even more effective.

**Effectiveness of *Rethink*** The goal of *rethink* is to improve the quality of the thought search by refining error thoughts, thereby enhancing the search quality within the same number of rollouts. To validate the effectiveness of *rethink*, we compare the performance between increasing the number of *rethink* operations and increasing the number of rollouts without applying *rethink*, while keeping the total number of rollouts consistent. The results are shown in Figure 5.

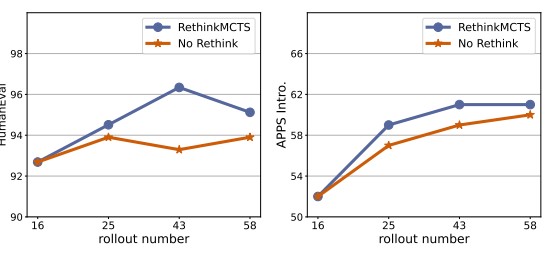

Figure 5: Performance comparison between *rethink* more times and more rollouts without *rethink*. *rethink* is more effective than increasing rollouts.

The figure shows that both increasing the number of *rethink* operations and increasing the number of rollouts both enhance the performance of code generation. This is expected as more extensive exploration raises the probability of finding the correct code. However, increasing *rethink* times is better than simply increasing rollouts. From a tree search perspective, without the *rethink* mechanism, erroneous thoughts would persist in the trace, causing the following nodes to follow incorrect reasoning paths, which makes it challenging to ensure the quality of the entire reasoning trace. The *rethink* mechanism refines these flawed thoughts based on the code feedback, enabling subsequent exploration along a better path and thus improving the overall quality of the tree search.

| Method | APPS Intro. | HumanEval |
|---|---|---|
| W/O RETHINK | 10.04 | 48.30 |
| RETHINKMCTS | **15.60** | **53.29** |

Table 3: The success rate comparison of the searched codes between with and without the *rethink* mechanism.

Furthermore, we compare the pass rate on public test cases of all the generated codes for the entire MCTS tree, with and without the *rethink* operation, since only public test cases are available during the search. The results are presented in Table 3. We can see that the *rethink* operation increases the proportion of effective code found in the tree. This highlights how refining erroneous thoughts enables the tree to focus more on correct paths, leading to better outcomes.

| $(a, b)$ | Pass Rate (%) | | | Pass@1 (%) | | | |
|---|---|---|---|---|---|---|---|
| | APPS Intro. | APPS Inter. | APPS Comp. | APPS Intro. | APPS Inter. | APPS Comp. | HumanEval |
| (0.8, 0.2) | 76.6 | 74.3 | 42.5 | 59 | 49 | **28** | **94.5** |
| (1.0, 0.2) | 76.9 | **76.4** | **43.5** | **60** | 53 | 27 | 92.7 |
| (1.0, 1.0) | **78.8** | 75.2 | 40.5 | **60** | **54** | 24 | 91.5 |

Table 4: Performance comparison under different reward weights. The (1.0, 0.2) and (1.0, 1.0) configurations make the nodes that achieve a pass rate of 1.0 on public test cases receive a score higher than 1.0, whereas the (0.8, 0.2) configuration keeps all node evaluations between 0∼1.

**Self-evaluation vs. Self-generating Unit Tests**  Given the limited coverage of public test cases, we propose a dual evaluation approach. Once the code passes all public test cases, we supplement it with a comprehensive self-evaluation score of the code from the LLM. In this section, we compare self-evaluation with the alternative approach of self-generating unit tests. In the latter approach, when the code passes the public test cases, we have the LLM generate additional test cases and get a new *pass rate* on these tests. The combined results serve as a comprehensive evaluation of the code. Experimental results are shown in Table 2.

As the table demonstrates, while self-generating unit tests improve the *pass rate* on test cases, they do not improve the *pass@1* metric. This is because self-evaluation directly assesses the code after it passes the public test cases, scoring it based on how well it meets the problem's requirements. As a result, it provides a more accurate indication of the code's ability to address the entire problem. In contrast, self-generating unit tests focus on creating additional tests, which emphasize the test suite rather than the code itself. There are two potential reasons for this: 1) Self-generating unit tests primarily identify patterns in the existing tests and generate a set of tests that better match the test suite. This can enhance the *pass rate* by filtering for code that matches these patterns, but it doesn't necessarily identify the mismatch between the code and the problem requirement. 2) The generated tests may not always be correct (Huang et al., 2023b), which can mislead the code's modification process and the subsequent search direction, potentially steering it away from valid solutions.

**Study on Reward Weights**  We conduct experiments to investigate the impact of reward weights in Eq. (3) of Sec. 3. The results are shown in Table 4. It is evident that $(a, b)$ significantly impacts the performance of RETHINKMCTS, highlighting the importance of LLM self-evaluation. The self-evaluation reward is only applied when the code achieves a pass rate of 1 on public test cases, so these different configurations have distinct implications. With the (0.8, 0.2) configuration, the code is given a baseline score of 0.8, and the LLM's evaluation score is used to distinguish between different codes. This allows for situations where the total score of code that passes all public test cases could be lower than that of code with a pass rate below 1, but only when the LLM's self-evaluation score is particularly low. Here, the LLM's self-evaluation score supplements the incomplete evaluation of public test cases. The main goal of the configurations (1.0, 0.2) and (1.0, 1.0) is to ensure that the overall score of codes with a pass rate of 1 is higher than 1. The advantage of this approach is that the final output code will most likely maintain a pass rate of 1 on public test cases, leading to better overall performance on the APPS dataset. However, since RETHINKMCTS is based on tree search, a node with a pass rate of 1 on public test cases doesn't always indicate a path worth exploring. Conversely, nodes with a pass rate below 1 may still warrant further exploration if their reasoning process is promising. Unfortunately, the (1.0, 0.2) and (1.0, 1.0) weights prematurely discard such paths, leading to poorer performance on the HumanEval and APPS Competition datasets.

# 6 RELATED WORK

**LLMs for Code Generation**  Large language models (LLMs) have been widely applied and developed in the field of code (Nam et al., 2024; Huang et al., 2023a; Li et al., 2024; He et al., 2024). Research on LLMs for code generation can be broadly divided into two categories: The first category focuses on fine-tuning LLMs, specifically on code data (Luo et al., 2023; Li et al., 2023; Fried et al., 2022; Roziere et al., 2023), which makes them to get a deep understanding of code syntax, semantics, and structures. Strong foundational models, like GPT-3.5-turbo and GPT-4, have also demonstrated impressive performance on code generation tasks due to their advanced pre-training on code data (Madaan et al., 2024). The second category is to use LLMs as agents (Ishibashi &

Nishimura, 2024; Zhang et al., 2024a; Jin et al., 2024). They usually design a procedure for generating codes and make LLMs play different roles. LDB proposed by Zhong et al. (2024) takes the LLM as a debugger and utilizes block-level decomposition to locate bugs, finally enhancing the code generation performance by iteratively debugging. PG-TD proposed by Zhang et al. (2023) utilizes Monte Carlo Tree Search (Browne et al., 2012) methods combined with the probabilistic output of LLMs to achieve token-level search for code generation. Reflexion proposed by Shinn et al. (2024) takes the LLM to generate reasoning, action, and reflections to make the LLM learn from past experience, which achieved impressive performance on code generation problems. Although they achieve great performance, these methods fail to build a process of devising the reasoning behind writing code for such a high-reasoning-demand task, which is the focus of our work.

**Tree Search-enhanced LLMs** Tree search methods can improve the reasoning performance of LLMs by exploring various possible paths (Wang et al., 2024; Meng et al., 2024; Yuan et al., 2024). By designing different action spaces, LLMs can explore at different levels (Zhang et al., 2023; Hu et al., 2024; Hao et al., 2023). PG-TD (Zhang et al., 2023) explores at the token level using the probability distribution of tokens from the Transformer (Vaswani, 2017) architecture, achieving fine-grained search for the correct code. Tree of Thoughts (ToT) (Yao et al., 2024) builds on the Chain-of-Thoughts (CoT) (Wei et al., 2022) by breaking down the steps of a task and exploring the thought for each step. With increasing research attention, more methods are being developed to better guide LLMs in performing tree search explorations. LATS (Zhou et al., 2023a) combines tree search with self-reflection. It searches at the code level for code generation and then summarizes the failures as reflections, storing them in memory for future generations. The error summarization is the verbal feedback of their method. Our approach takes a different path. Instead of merely summarizing or reflecting on errors, we employ block-level analysis as verbal feedback. This feedback is then used to regenerate MCTS nodes (refining erroneous thoughts), thus enhancing the overall quality of the trace by systematically refining incorrect reasoning within it. TS-LLM (Feng et al., 2023) proposes a tuning-based method that trains a value function to guide the decoding process. By contrast, we focus on tuning-free algorithms that enhance the code generation capabilities of LLMs in an off-the-shelf manner. While these methods successfully enhance the task-solving abilities of LLMs, they may not fully harness the potential of tree search in code generation tasks. This is largely because many of these approaches focus on token- or code-level searches, overlooking the deeper reasoning process that is critical for tasks like code generation, which require intricate reasoning. Additionally, the detailed execution feedback provided by the code environment has great potential to guide the search process, but these methods fall short of effectively integrating this feedback into the search. In this paper, we focus on leveraging detailed feedback from the code execution environment to guide and refine the thought process, thereby improving the overall quality of exploration.

## 7 CONCLUSION

In this work, we propose RETHINKMCTS, the first framework designed to search and refine thoughts for code generation. Unlike previous tree search-enhanced methods, RETHINKMCTS explores different coding strategies by navigating through the reasoning process and introduces a feedback-based refinement mechanism to enhance search quality. This approach effectively utilizes execution information to construct verbal feedback, thereby refining erroneous reasoning steps. Additionally, in the evaluation phase, we introduce a dual evaluation method to address the incomplete coverage of public test cases. Through comparative experiments on the APPS and HumanEval datasets, we demonstrate that RETHINKMCTS achieves the best performance, demonstrating its ability to effectively generate high-quality code by searching and refining the reasoning process. Beyond code generation tasks, RETHINKMCTS provides a general approach for enhancing task performance through a search-and-refinement reasoning process, making it have the potential to be applied to other domains handled by LLM agents, such as mathematical problem-solving and tool usage scenarios.

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

## APPENDIX

## A    ADDITIONAL RESULTS

This section presents some additional experiment results.

**Ablation Study**    Here, we present the results of the ablation study using GPT-4o-mini as the backbone model, as shown in Figure 6. It is clear that the *rethink* operation and verbal feedback remain the most significant contributors to our model's performance. Notably, the *rethink* mechanism exhibits even stronger effects with GPT-4o-mini than with GPT-3.5-turbo, likely due to the model's enhanced ability to effectively utilize feedback and make refinement.

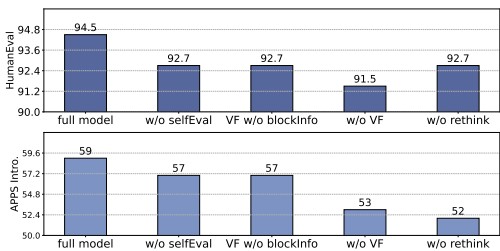

Figure 6: Ablation study of block-level analysis (blockInfo), rethink mechanism, the verbal feedback (VF) and self-evaluation with GPT-4o-mini as the backbone.

Figure 7: Performance comparison between different search granularity. For advanced model like GPT-4o-mini, it's better to explore at the thought level.

**Search Granularity Study**    We present the results of the search granularity study using GPT-4o-mini as the backbone model, shown in Figure 7. It is evident that the differences across granularities are smaller on the HumanEval dataset, likely due to its relatively low overall difficulty. However, on the APPS dataset, the advantage of thought-level search becomes much more pronounced, especially at the highest "competition" difficulty level. This suggests that for more complex problems, exploring the thought process and reasoning is beneficial.

## B    PROMPTS

In this section, we present the prompts used when an LLM acts as an agent to perform various operations.

### B.1    EXPANSION PROMPT

First, we discuss the prompts for the Expansion step in the MCTS process. There are two sets of prompts: one set is used to generate new thoughts based on the problem description and previous thoughts when there is no feedback;

```
{problem_statement}

{thoughts}
```

```
Above is a problem to be solved by Python program.

I need you analyze this problem and provide strategies. I need you to output {width} possible thoughts and strategies.

Remember each only contain one possible strategy of the problem. Please wrap your response into a JSON object that contains
keys 'Thought-i' with i as the number of your thought, and key 'Reasonableness' with the Reasonableness of each thought,
which should between 0~1 and the sum should be 1.

The JSON should be a ** list of dicts **, the dicts are splited with comma ','.
Example Answers:
[
    {"Thought-1":" We could use the print function to finish the task in one line: print(2 + 3)", "Reasonableness": 0.7},
    {"Thought-2":" We should calculate the problem by setting a=2+3, and then print(a)", "Reasonableness": 0.29},
    {"Thought-3":" The problem can't be solved by Python.", "Reasonableness": 0.01}
]
```

The other set is used when the generated code contains errors and verbal feedback is provided. In this case, the LLM uses the verbal feedback to generate thoughts that avoid such errors.

```
{problem_statement}

{thoughts}

``` python
{generated_code}
```

{verbal feedback}

I need you to analyze and provide new thoughts that can lead to the correct solution code.

The goal is that the thoughts could lead to the code that not only avoids the current error but also solve the problem in a
way that handles other potential test cases that we haven't encountered yet. I need you to output {self.width} possible
thoughts and strategies. Remember each only contain one possible strategy of the problem.

Please wrap your response into a JSON object that contains keys 'Thought-i' with i as the number of your thought, and key '
Reasonableness' with the Reasonableness of each thought, which should between 0~1 and the sum should be 1.

The JSON should be a ** list of dicts **, the dicts are splited with comma ','.

Example Answers:
[
    {"Thought-1":" We could use the print function to finish the task in one line: print(2 + 3)", "Reasonableness": 0.7},
    {"Thought-2":" We should calculate the problem by setting a=2+3, and then print(a)", "Reasonableness": 0.29},
    {"Thought-3":" The problem can't be solved by Python.", "Reasonableness": 0.01}
]
```

## B.2 CODE GENERATION PROMPT

We present the prompt we use to instruct the LLM to generate code following previous thoughts.

```
{problem_statement}

{thoughts}

Complete the Python program to solve the problem. Remember to contain the complete program including all the imports and
function header in your response.

Also some thoughts are included that you can refer to and build upon when writing the code.

Answer with the code ONLY. No other explanation or words attached !
```

## B.3 EVALUATION PROMPT

Besides the normal evaluation on the public test cases, we also develop an LLM-based self-evaluation when the public test cases are all passed. Here we present the prompts.

```
{problem_statement}
```

```
{thoughts}

``` python
{generated_code}
```
Above is a Python code problem with the thoughts and code to solve the problem. The code could pass all the example test
cases, however, it may or may not be completely correct.

Please evaluate and return the correctness score in range [−1, 1].

Evaluate the correctness of the code and give only ONE evaluation score.

The code's correctness is whether it can pass all the possible unseen test cases of the problem, not just the given ones.

Example Answers:
{"evaluation": −0.5, "explanation": The code is far from correct for solving the problem.}
{"evaluation": 0.1, "explanation": The code is not the correct solution but can pass some simple test cases.}
{"evaluation": 0.85, "explanation": The code can pass most test cases while may fail on some corner cases.}
{"evaluation": 1.0, "explanation": The generated code is the correct solution that can pass all the possible test cases and
strange corner cases too.}
```

## B.4 RETHINK PROMPT

When the generated code following some thoughts doesn't pass some public test cases, we would
use the block-level analysis to form the verbal feedback and use it to refine the previous thought,
a.k.a, rethink. Here we present the prompt for this operation.

```
{problem_statement}

{thoughts}

``` python
{generated_code}
```

{verbal feedback}

Based on your previous thoughts and the new experience, please provide a new Thought to replace the previous thought. This
new thought should avoid the mistake.

Remember that you only need to provide the thought (one or two sentences) to solve the problem, not the code.
```

## B.5 AN EXAMPLE OF VERBAL FEEDBACK

The verbal feedback we constructed contains the detailed block-level analysis of the code. Here we
present an example of it.

```
(Problem and Code)
import math
def largest_divisor(n: int) −> int:
    """ For a given number n, find the largest number that divides n evenly, smaller than n
    >>> largest_divisor(15)
    5
    """
    for i in range(int(math.sqrt(n)), 0, −1):
        if n % i == 0 and i < n:
            return i
    return 1  # In case n is 1, return 1 as the largest divisor

(Execution Feedback)
The code above fails the given unit test:
assert largest_divisor(15) == 5
# Current Execution Output:
3.

(Block−level Analysis)
Here is the code execution trace block by block with the intermediate variable values. Please explain the execution FOR
EACH BLOCK and answer whether this block is correct or not. If not, give an explanation on what is wrong. Please wrap your
response into a JSON object that contains keys 'block' with the name of each block, key 'correct' with value False or True,
and key 'explanation' with an explanation on the bug.
Example Answers:
{"block": "BLOCK−1", "correct": "True", "explanation": "The block initializes variable 'a' and 'b'."}
```

```
{"block": "BLOCK−2", "correct": "False", " explanation ": "The block is incorrect because the code does not add the two
 integers together, but instead subtracts the second integer from the first. To fix this issue, we should change the operator
 from '−' to '+' in the return statement. This will ensure that the function returns the correct output for the given input
."}
[BLOCK−0]
    # n=15
    for i in range( int (math. sqrt (n)), 0, −1):
    # n=15  i=3
[BLOCK−1]
        # n=15  i=3
        if n % i == 0 and i < n:
        # n=15  i=3
[BLOCK−2]
            # n=15  i=3
            return i
            # n=15  i=3  ret =3

# LLM−analysis
``` json
[
    {"block": "BLOCK−0", "correct": "True", " explanation ": "The block  initializes  the loop with the correct  starting  point,
    which is the integer  part of the square  root of n."},
    {"block": "BLOCK−1", "correct": "False", " explanation ": "The block checks if 3 is a divisor of 15, which is correct, but
     we are not immediately returning the  largest  divisor. The loop continues, and the next iteration will check smaller
    values of i ."},
    {"block": "BLOCK−2", "correct": "False", " explanation ": "The block incorrectly  returns 3 instead of the larger  valid
    divisor. Since the loop searches  from the square  root downwards, it should continue checking until  it finds the next
    valid divisor, which is 5. The return  statement needs to occur outside of this loop and only when the largest  divisor  is
     found."}
]
```
```

## C ALGORITHM

We present the detailed procedure of RETHINKMCTS in pseudocode in Algorithm 1.

**Algorithm 1** The RETHINKMCTS algorithm.

---

**Require:** $root$: the problem description; $c$: P-UCB exploration parameter; $k$: the maximum number of children of any node; $a, b$: the reward weights of the pass rate and the LLM evaluation.

1: $program\_dict = \text{DICTIONARY}()$
2: $verbal\ feedback\ f = \text{EMPTY}$
3: **for** $i \leftarrow 1, 2, \ldots, max\_rollouts$ **do**
4:     $node \leftarrow root$
5:     # Selection
6:     **while** $|node.children| > 0$ **do**
7:         $node \leftarrow \text{P\_UCB\_SELECT}(node.children, c)$
8:     **end while**
9:     # Expansion
10:     $next\_thoughts \leftarrow \text{TOP\_K}(node, k)$
11:     **for** $next\_thought \in next\_thoughts$ **do**
12:         $next\_state \leftarrow \text{CONCAT}(node, next\_thought)$
13:         Create a node $new\_node$ for $next\_state$
14:         Add $new\_node$ to the children of $node$
15:     **end for**
16:     # Evaluation
17:     $C \leftarrow \text{GENERATE}(node)$
18:     $v^{\text{test}}, f \leftarrow \text{GET\_PASS\_RATE}(p)$
19:     $v^{\text{llm}}, f \leftarrow \text{GET\_LLM\_EVAL}(p)$
20:     $program\_dict[C] = r = a * v^{\text{test}} + b * v^{\text{llm}}$
21:     **if** $v^{\text{test}} = 1$ **then**
22:         $program\_dict[C] = r = a * v^{\text{test}} + b * v^{\text{llm}}$
23:     **else**
24:         $program\_dict[C] = r = v^{\text{test}}$
25:     **end if**
26:     # Backpropagation
27:     Update and the values of $node$ and its ancestors in the tree with $r$
28:     # Rethink
29:     **if** $v^{\text{test}} \neq 1$ **then**
30:         $node.thought = \text{RETHINK}(node, f)$
31:         $next\_thoughts = \text{RETHINK\_NEXT}(node, k, f)$
32:         $C = \text{RE-GENERATE}(node)$
33:         $r = \text{RE-EVALUATION}(C)$
34:         $program\_dict[C] = r$
35:     **end if**
36: **end for**
37: **return** program in $program\_dict$ with the highest reward

---

