# OpenReview forum: "RethinkMCTS: Refining Erroneous Thoughts in Monte Carlo Tree Search for Code Generation"
_ICLR.cc/2025/Conference — Submitted to ICLR 2025_

### Official Review · Reviewer_GpRr · 2024-10-25

**Soundness:** 3
**Presentation:** 3
**Contribution:** 3
**Rating:** 6
**Confidence:** 4

**Summary:**

This paper proposes to use Monte-Carlo tree search to discover useful rationale for code generation. It also incorporates block-level info feedback and self-evaluation for better value estimate of nodes in the tree. Experiment results show that the proposed method achieves SOTA performance compared to other augmented code generation method, such as reflexion and tree-of-thoughts.

**Strengths:**

The propose method is novel and provides a nice way to combine MCTS and chain-of-thoughts. The paper is well-written and easy to understand.

**Weaknesses:**

The experiment results can be further strengthen:
1. for such a searching algorithm, the results between different runs might differ a lot. Thus, it is better to incorporate std information for pass rate.
2. The run time of different algorithms should also be compared.

**Questions:**

Nothing necessary stands out.

---

> ### Author Response · Authors · 2024-11-21
> **Author Response**
>
> Thank you for dedicating your time and effort to our work.
>
> > For such a searching algorithm, the results between different runs might differ a lot. Thus, it is better to incorporate std information for pass rate.
>
> We conducted three experiments comparing our method with competitive baselines. The mean and standard deviation of the results are presented below:
> |Model|APPS-intro.(Pass Rate %)|APPS-inter.(Pass Rate %) |APPS-comp.(Pass Rate %)|APPS-intro.(Pass@1)|APPS-inter. (Pass@1)|APPS-comp.(Pass@1)|Humaneval (Pass@1)|
> |-|-|-|-|-|-|-|-|
> |**(GPT4o-mini)**||||||||
> | ToT| 75.11 $\pm$ 0.57|71.79 $\pm$ 0.36|42.5 $\pm$ 0.54|56.33 $\pm$ 1.89|48.33 $\pm$ 1.25|25.33 $\pm$ 1.7|93.5 $\pm$ 0.29|
> | LATS | 70.17 $\pm$ 0.52|68.66 $\pm$ 0.79|36.83 $\pm$ 3.61|50.33 $\pm$ 0.47|45.67 $\pm$ 0.94|18.33 $\pm$ 3.3|93.7 $\pm$ 0.57|
> | PG-TD | 68.85 $\pm$ 2.29|68.29 $\pm$ 1.48|40.33 $\pm$ 2.00|49 $\pm$ 4.32|44.33 $\pm$ 1.25|23 $\pm$ 1.63|92.28 $\pm$ 0.76|
> | RethinkMCTS| **75.36**$\pm$ 1.08 | **74.10** $\pm$ 0.98| **43.33** $\pm$ 1.18|**57.33** $\pm$ 1.25|**50** $\pm$ 0.82|**27** $\pm$ 0.82|**94.31** $\pm$ 0.29|
>
> Noticeably, our model, RethinkMCTS, consistently demonstrates a stable performance advantage across multiple experiments.
>
> > The run time of different algorithms should also be compared.
>
> Here, we present a comparison of the inference time between our method and several major tree search baselines.
>
> |Model|APPS-intro.|Humaneval|
> |-|-|-|
> | **(GPT4o-mini)**|||||
> | RethinkMCTS|344.18s | 160.87s|
> | LATS |207.83s | 129.64s|
> | ToT| 117.37s | 92.93s|
> | PG-TD |113.44s | 99.5s|
>
> Our method indeed requires more time than other baselines. However, it is essential to note that, on average, tree search algorithms take significant time to solve each problem. This is because these methods aim to trade time for improved results, embodying the "slow thinking" approach to reasoning. For instance, OpenAI's o1 model[1] can also take 3-5 minutes to solve a single problem.
>
> [1] https://openai.com/index/learning-to-reason-with-llms/

---

> ### Author Response · Authors · 2024-11-25
> **Official Comment by Authors**
>
> Dear Reviewer,
>
> Thank you once again for your insightful comments and valuable suggestions.
>
> We hope that our response has adequately addressed your concerns. If you have any further questions or feedback, please don't hesitate to let us know. We are committed to actively engaging with your input and continuously improving our paper.
>
> We look forward to receiving your further comments.
>
> Best regards,
>
> The Authors

---

> > ### Comment · Reviewer_GpRr · 2024-11-30
> > **Response to rebuttal**
> >
> > Thanks for the responses to my questions. I have no other concerns.

---

### Official Review · Reviewer_VEZu · 2024-10-31

**Soundness:** 2
**Presentation:** 3
**Contribution:** 3
**Rating:** 6
**Confidence:** 5

**Summary:**

This paper proposes an MCTS-based approach to incorporate reasoning into code generation. A key hypothesis is that searching in the space of thoughts is better than searching in the space of code, and that searching in thought space allows the LLM to explore better as well as have a higher quality of search by refining its intermediate outputs. The feedback used in MCTS is a combination of scalar feedback and verbal feedback generated by the LLM. This approach differs from other reflection-based approaches using tree search in that intermediate nodes in the tree are refined so that the LLM can follow a better path to the solution using the reasoning provided in the form of verbal feedback. This work also proposes combining LLM self-evaluations with public test cases to evaluate the generated code, to compensate for the low coverage of public test cases with respect to the problem, leading to inaccurate assessment of the generated solution. The results of the proposed RethinkMCTS approach are demonstrated on the HumanEval and Apps datasets, for GPT-3.5-turbo and GPT-4o-mini.

**Strengths:**

- The paper proposes a neat, conceptually simple way of doing MCTS in the space of LLMs for code generation.
- This work suggests an alternative to current approaches to handle incomplete coverage of solutions by public test cases by proposing LLM self-assessment instead of generating synthetic test cases that may not always be accurate.
- The method section is written well, clearly outlining the different parts of the MCTS algorithm and what they correspond to in this context, as well as the additional verbal feedback and rethink stages that are the novel contributions of the proposed approach.
- The ablation results in Figure 3 and Figure 6, and the search granularity results in Figure 4 and Figure 7 tie together each individual contribution in the paper with the main rethink hypothesis in the paper.
- The proposed approach is compared with several feedback based and tree-search based baselines.

**Weaknesses:**

- In Table 1, the RethinkMCTS results have been marked in bold, which I presume indicates the superior performance of this method over other baselines. However, it seems to be the case that for GPT-4o-mini, the results shown by the Tree-of-Thought baseline are in most cases, at par with RethinkMCTS, and the pass rate of PG-TD on APPS-Comp is actually higher than that of RethinkMCTS. This table would be clearer to read if the best baselines that are at par with or better than RethinkMCTS were also highlighted.
- The absence of standard errors in Table 1 makes it difficult to see if the performance improvements are significant or not.
- One important detail that would warrant some discussion in this paper is the number of LLM calls used to generate the final output. Ideally for a fair comparison with baselines, this would be an important parameter to standardise, since recent work in inference-time approaches have shown that given more compute during inference, LLMs can exhibit superior performance. If the baselines use fewer LLM calls at inference time than RethinkMCTS to generate the final output, this would not be a fair comparison.
- The choice of datasets and models makes the evaluation of this approach difficult: given that the base models already show strong performance on these datasets, it becomes difficult to assess the contribution of this approach, which has several components, in increasing performance. If this approach shows significant gains in performance, for either a) a more difficult dataset, or b) weaker models, that might make the strength of the approach as well as its individual components clearer.
- Minor: There is an overloading of the term $a$, in some places it is used to denote the action, in others it is used as a coefficient for the reward weight vector.

**Questions:**

- The improvements shown by RethinkMCTS compared to other baselines seem to be more pronounced on GPT-3.5-turbo compared to GPT-4o-mini. Is there anything that can be inferred from this result that can tie the effectiveness of this approach with the capabilities of the base model?
- In Figure 3, the result on HumanEval indicates that the rethink mechanism seems to have the least effect out of all components on performance. For all of the other results as well (except APPS Intro. in Figure 6), the rethink component is not the most impactful in determining performance. This appears to be contrary to the claim of this paper. Could the authors clarify how these results were interpreted?
- In Figure 5(a), as the number of rollouts increases from 43 to 58, the performance with rethink reduces, which is counterintuitive. The result in 5(b) is more consistent with what we could expect as we increase the number of rollouts: performance would plateau beyond a certain number of rollouts. Could the authors provide an explanation for why we see the drop in performance in Figure 5(a)?
- The numbers in Table 3 are very different from those in Figure 3; is it a different metric, model, or dataset?
- GPT-4o-mini has comparable token-level search performance compared to RethinkMCTS on HumanEval; is it possible that HumanEval is not the best testbed for this approach, given that base performance is already quite high?

---

> ### Author Response · Authors · 2024-11-21
> **Author Response (1/3)**
>
> Thank you for dedicating your time and effort to our work.
>
> > The pass rate of PG-TD on APPS-Comp is higher.
>
> > Table 1 would be clearer to read if the best baselines that are at par with or better than RethinkMCTS were also highlighted.
>
> > There should be standard errors in Table 1.
>
> It's our fault that the pass rate of PG-TD at the competition difficulty level was not bolded. We will correct this in the revision. Additionally, we will underline the second-best row in the table in the revised version to improve clarity. (We tried but openreivew's editor does not support underline, so we would not put the underline in the table below.)
>
> Here, we ran each experiment involving GPT-4o-mini from the main table three times. The mean and std are presented below:
> |Model|APPS-intro.(Pass Rate %)|APPS-inter.(Pass Rate %) |APPS-comp.(Pass Rate %)|APPS-intro.(Pass@1)|APPS-inter.(Pass@1)|APPS-comp.(Pass@1)|HumanEval (Pass@1)|
> |-|-|-|-|-|-|-|-|
> | **(GPT4o-mini)**||||||||
> | ToT| 75.11 $\pm$ 0.57|71.79 $\pm$ 0.36|42.5 $\pm$ 0.54|56.33 $\pm$ 1.89|48.33 $\pm$ 1.25|25.33 $\pm$ 1.7|93.5 $\pm$ 0.29|
> | LATS | 70.17 $\pm$ 0.52|68.66 $\pm$ 0.79|36.83 $\pm$ 3.61|50.33 $\pm$ 0.47|45.67 $\pm$ 0.94|18.33 $\pm$ 3.3|93.7 $\pm$ 0.57|
> | PG-TD | 68.85 $\pm$ 2.29|68.29 $\pm$ 1.48|40.33 $\pm$ 2.00|49 $\pm$ 4.32|44.33 $\pm$ 1.25|23 $\pm$ 1.63|92.28 $\pm$ 0.76|
> | RethinkMCTS| **75.36**$\pm$ 1.08 | **74.10** $\pm$ 0.98| **43.33** $\pm$ 1.18|**57.33** $\pm$ 1.25|**50** $\pm$ 0.82|**27** $\pm$ 0.82|**94.31** $\pm$ 0.29|
>
>
> We want to point out that in code generation experiments, it is common not to compare std[1, 2] after setting the temperature to 0. The variations in LLM outputs are minimal under his setting.
>
> > If the baselines use fewer LLM calls at inference time than RethinkMCTS to generate the final output, this would not be a fair comparison.
>
> First, we want to clarify that the goal of using tree search algorithms to enhance reasoning is to enable LLMs to engage in "slow thinking." Like in OpenAI's o1[3] model, the processing time and token consumption are not the goal of "slow thinking" (o1 could solve a simple problem with thousands of tokens in the hidden CoT and minutes to take [3]).
>
> Our core contribution lies in focusing on the reasoning process in code generation tasks. This perspective had yet to be explored before OpenAI's o1 model (our work was conducted concurrently with o1). We propose a "rethink" mechanism to refine the thought process of writing code. We believe this introduces a new research focus for code generation tasks: emphasizing the thought process behind writing code rather than merely aiming to search for the correct codes simply.
>
> The way to keep a fair comparison in tree search works is to keep the number of codes generated the same [2,4] (i.e., the rollout number in our paper). Regarding the number of LLM calls, RethinkMCTS involves additional calls for "rethink" during each rollout compared to ToT. However, it does not increase the number of LLM calls compared to LATS (MCTS + reflection), as the latter uses extra calls for summarizing reflections, whereas we use them for "rethink." Experiments demonstrate that our "rethink" mechanism is more effective.
>
> [1] Shinn, N., et al. Reflexion: Language agents with verbal reinforcement learning. NIPS 2024.
>
> [2] Zhang, S., et al. Planning with large language models for code generation. ICLR 2023.
>
> [3] https://openai.com/index/learning-to-reason-with-llms/
>
> [4] Zhou, A., et al. Language agent tree search unifies reasoning acting and planning in language models. PMLR 2024.

---

> ### Author Response · Authors · 2024-11-21
> **Author Response (2/3)**
>
> > If this approach shows significant gains in performance, for either a) a more difficult dataset, or b) weaker models, that might make the strength of the approach as well as its individual components clearer.
>
> 1. We chose GPT-3.5 and GPT-4o-mini to compare a stronger reasoning model with a weaker one while keeping experimental costs reasonable. We did not choose smaller open-source models due to their limited reasoning capabilities. Similar to the "slow thinking" approach of OpenAI's o1 model, our enhancement methods aim to improve the reasoning performance of models already possessing some reasoning ability. Thus, a baseline model with sufficient capabilities is necessary, as pointed out in [5].
> 2. The APPS dataset offers a comprehensive range of problems with a broad difficulty distribution, featuring over 1,000 questions at each difficulty level. However, due to constraints on experimental costs, we initially tested only the first 100 questions at each difficulty level. In response to your concerns about performance comparisons, we have expanded the experimental scope to include 200 questions per difficulty level on GPT-4o-mini, allowing for a more thorough comparison across multiple models.
>
> |Model|APPS-intro.(Pass Rate %)|APPS-inter.(Pass Rate %) |APPS-comp.(Pass Rate %)|APPS-intro.(Pass@1, %)|APPS-inter.(Pass@1, %)|APPS-comp.(Pass@1, %)|
> |-|-|-|-|-|-|-|
> | **(GPT4o-mini)**|||||||
> |ToT|77.5451  |72.7576 |46.3333 |62 |49.5 |30.5 |
> |LATS| 74.0187 |69.6501 |45.8333 |57 |47 |30 |
> |PG-TD| 69.2456 |68.5919 |44.6667 |52.5 |46 |29.5 |
> |RethinkMCTS|**79.5234** |**75.1658** |**50** |**66.5** |**53** |**32** |
>
> We can see that the advantage is more obvious when the number of questions rises.
>
> > Minor: There is an overloading of the term a.
>
> Thank you very much for your feedback. We have noted the issue you mentioned and will use a different, non-repeating letter to represent the coefficient.
>
> > The improvements shown by RethinkMCTS compared to other baselines seem to be more pronounced on GPT-3.5-turbo compared to GPT-4o-mini. Is there anything that can be inferred from this result that can tie the effectiveness of this approach with the capabilities of the base model?
>
> 1. Although RethinkMCTS shows more pronounced effects on GPT-3.5, the effectiveness of searching at the thought level, rather than focusing on code or tokens as other methods do, is evident in both base models. This supports our contribution of emphasizing the reasoning process behind the code.
> 2. As for whether it is more effective on weaker models, if a model has strong inherent reasoning capabilities (like 4o-mini), it may already perform sufficient reasoning when generating responses, making the improvements marginal. However, we do not believe that search methods are suitable for very small, weak models, as forcing them into complex reasoning might lead to incorrect or misleading reasoning and, therefore, might not be appropriate.
>
> > In Figure 3, the rethink mechanism seems not much impactful in determining performance.  Could the authors clarify how these results were interpreted?
>
> 1. Verbal feedback forms the foundation of the "rethink" mechanism. Previous research has demonstrated that LLMs lack the ability to self-correct without external feedback[6]. We introduced the "rethink" mechanism to leverage the rich feedback available in coding environments; thus, verbal feedback may seem more pronounced in ablation.
> 2. On HumanEval, the model's initial pass rate is relatively high, limiting the opportunities for "rethink" operations since "rethink" is only executed when the initial code generated for the current node is incorrect.
> 3. Figure 6 in Appendix A further explores ablation on GPT-4o-mini. While the effect of "rethink" on HumanEval is similar to verbal feedback, it has a much greater impact on APPS, where the initial pass rate is lower. This aligns with the findings in Figure 3, suggesting that the limited impact on HumanEval is due to fewer opportunities for "rethink" to be utilized effectively.
>
> [1] Shinn, N., et al. Reflexion: Language agents with verbal reinforcement learning. NIPS 2024.
>
> [2] Zhang, S., et al. Planning with large language models for code generation. ICLR 2023.
>
> [3] https://openai.com/index/learning-to-reason-with-llms/
>
> [4] Zhou, A., et al. Language agent tree search unifies reasoning acting and planning in language models. PMLR 2024.
>
> [5] Li, J., et al. CodeTree: Agent-guided Tree Search for Code Generation with Large Language Models. arXiv preprint arXiv:2411.04329.
>
> [6] Huang, J., et al. Large language models cannot self-correct reasoning yet. arXiv preprint arXiv:2310.01798.

---

> ### Author Response · Authors · 2024-11-21
> **Author Response (3/3)**
>
> > In Figure 5(a), as the number of rollouts increases from 43 to 58, the performance with rethink reduces, which is counterintuitive. Could the authors provide an explanation?
>
> 1. A key limitation lies in the evaluation stage. While the search algorithm generates multiple candidate codes, only one is ultimately selected for testing based on the pass rate on public test cases. However, the limited number of public test cases can lead to inaccuracies. For example, if multiple codes pass all public tests, one is selected at random, but it’s possible that the chosen code contains errors. As rollouts increase, the likelihood of erroneous codes passing all public test cases also rises. Despite using LLM evaluation as a secondary method, this margin of error remains, so more rollouts do not necessarily improve results.
> 2. Another possible explanation is that the model has reached its performance ceiling. Further searching may not generate better code, and the observed decline in performance could be due to evaluation inaccuracies rather than actual limitations in the search process.
>
> > The numbers in Table 3 are very different from those in Figure 3; is it a different metric, model, or dataset?
>
> The metrics in Figure 3 represent the success rate of the final code generated by the tree search across 100 problems. The values in Table 3 are the proportion of correct codes found within one search tree. Table 3 aims to demonstrate that search trees incorporating the "rethink" mechanism discover more correct codes.
>
> > GPT-4o-mini has comparable token-level search performance compared to RethinkMCTS on HumanEval; is it possible that HumanEval is not the best testbed for this approach, given that base performance is already quite high?
>
> HumanEval is indeed less effective for distinguishing differences in performance with models like GPT-4o-mini due to their already high baseline performance. Thus, our primary comparisons for GPT4o-mini are focused on the three difficulty levels of the APPS dataset.
>
> However, we include HumanEval because it remains a standard benchmark in the code generation field [1, 4] and is effective for highlighting algorithmic differences in models like GPT-3.5, where there is still significant room for improvement.
>
> [1] Shinn, N., et al. Reflexion: Language agents with verbal reinforcement learning. NIPS 2024.
>
> [2] Zhang, S., et al. Planning with large language models for code generation. ICLR 2023.
>
> [3] https://openai.com/index/learning-to-reason-with-llms/
>
> [4] Zhou, A., et al. Language agent tree search unifies reasoning acting and planning in language models. PMLR 2024.
>
> [5] Li, J., et al. CodeTree: Agent-guided Tree Search for Code Generation with Large Language Models. arXiv preprint arXiv:2411.04329.
>
> [6] Huang, J., et al. Large language models cannot self-correct reasoning yet. arXiv preprint arXiv:2310.01798.

---

> ### Author Response · Authors · 2024-11-25
> **Official Comment by Authors**
>
> Dear Reviewer,
>
> Thank you once again for your insightful comments and valuable suggestions.
>
> We hope that our response has adequately addressed your concerns. If you have any further questions or feedback, please don't hesitate to let us know. We are committed to actively engaging with your input and continuously improving our paper.
>
> We look forward to receiving your further comments.
>
> Best regards,
>
> The Authors

---

> > ### Comment · Reviewer_VEZu · 2024-11-30
> >
> > I thank the authors for responding to the questions and concerns raised, and for the clarifications provided. I will keep my score for the following reasons:
> >
> > > we do not believe that search methods are suitable for very small, weak models, as forcing them into complex reasoning might lead to incorrect or misleading reasoning and, therefore, might not be appropriate
> >
> > This point would have more impact if the authors could point to existing results with either rethink or other mechanisms that show that other models that are not as capable as GPT-3.5 or GPT-4o-mini at reasoning would fail at search-based methods.
> >
> > > HumanEval is indeed less effective for distinguishing differences in performance with models like GPT-4o-mini due to their already high baseline performance. Thus, our primary comparisons for GPT4o-mini are focused on the three difficulty levels of the APPS dataset.
> >
> > If HumanEval is less effective for assessing the strength of the approach, there should be an alternative dataset for code generation to show the effectiveness of the approach (LiveCodeBench, CodeContests, to name a few). Or alternatively, as I suggested earlier, using a less competent model than the ones considered, which do not have high initial performance on HumanEval, might have made the point better.
> >
> > > The way to keep a fair comparison in tree search works is to keep the number of codes generated the same.
> >
> > I disagree with this point: if rethink uses several more intermediate LLM calls to then generate the final outputs, to have a fair comparison, the baselines should be allowed the same number of LLM calls at inference time as the full rethink algorithm.

---

### Official Review · Reviewer_vanH · 2024-11-01

**Soundness:** 2
**Presentation:** 3
**Contribution:** 2
**Rating:** 3
**Confidence:** 3

**Summary:**

In this paper, the authors propose an enhancement of Monte-Carlo Tree Search, named RethinkMCTS, that refines reasonings with feedback in code generation tasks. The search incorporates both execution feedback and verbal feedback to refine reasonings. Empirical evaluations on two coding benchmarks show modest pass metrics improvements over existing baselines.

**Strengths:**

1. This paper is well-written and easy to understand. The various feedback and the rethink process are integrated quite well.

2. The included baselines are comprehensive.

**Weaknesses:**

1. The novelty of this paper seems limited. I am struggling to understand the difference from the LATS and ToT work. In comparison, it seems like the proposed method is essentially LATS + ToT? If that is correct, I think the contribution is of limited significance.

2. The improvements over baselines, especially ToT, are small. With GPT-4o-mini, the differences were quite small. As there are no confidence intervals, there is doubt about the statistical significance of those improvements.

3. It is unclear to me whether the experiment results in Table 1 were controlled for the number of tokens across different methods, which is crucial for a fair comparison. Please provide more details about the comparison setting.

4. One part of the novelty is the block-level feedback, however, there is no example given for what this type of feedback looks like and no explanation about implementation details. Please provide more information.

**Questions:**

Please see the weakness section.

---

> ### Author Response · Authors · 2024-11-21
> **Author Response (1/2)**
>
> Thank you for dedicating your time and effort to our work.
>
> > I am struggling to understand the difference from the LATS and ToT work. In comparison, it seems like the proposed method is essentially LATS + ToT?
>
> Here, we would like to clarify the difference between RethinkMCTS, LATS, and ToT.
> 1. Difference from LATS: LATS combines tree search with reflection. As illustrated in Figure 1 of our paper, the rethink mechanism differs fundamentally from reflection. Reflection summarizes past errors into textual feedback, which is then appended to the state of subsequent nodes. In contrast, our rethink mechanism directly corrects erroneous nodes. This approach offers a significant advantage over reflection by removing incorrect nodes from the tree expansion process, enabling the search to follow higher-quality traces. As shown in Figure 5, our experiments confirm that as the number of rollouts increases (i.e., the tree expands further), the rethink mechanism effectively enhances search quality.
> 2. Difference from ToT: 1)The ToT tree search lacks feedback and refinement mechanisms, while our rethink mechanism significantly improves search quality by addressing these limitations. 2) Regarding the exploration of "thoughts": Although the ToT paper's title suggests a focus on searching "thoughts," its actual method involves decomposing tasks into steps and performing searches within each step. We adapted ToT's search to align with the code reasoning search approach used in RethinkMCTS, which resulted in strong performance. In another study of code generation AgentCoder[1], ToT performed poorly because the authors do not adopt ToT to code as we do.
>
> Why RethinkMCTS is not simply a combination of LATS and ToT: 1) The rethink mechanism is unique for both ToT and LATS. 2) Neither approach utilizes fine-grained block-level feedback to enhance tree search. 3) Our dual-evaluation method, which combines public test case evaluation with LLM-based scoring, is absent in either of these prior works.
>
> > The improvements over baselines, especially ToT, are small.
>
> About ToT baseline implementation.
> Here, we would like to address some details about the ToT baseline:
> 1. The search for reasoning/thought process for code is one of our key contributions. The strong performance of ToT is largely because we did not implement it strictly as described in the original paper. While "thoughts" are mentioned in its title, the original paper would decompose the task into fixed steps and search for solutions within each step. In our implementation, we merged ToT with our method by defining the search space in terms of reasoning "thoughts." This significantly improved its performance, making the ToT implementation in the paper an ablation of our method. The implementation approach likely plays a significant role in determining its performance. For instance, in the AgentCoder paper[1], ToT performs poorly, whereas when integrated with our searching reasoning process, it achieves much better results.
> 2. Our model demonstrates a clear advantage, especially on models like GPT-3.5, which have weaker coding capabilities. This advantage persists even in the rethink-only setting.
>
> To better illustrate our point, we strictly follow the paper Tree of Thought and adapt a task-decomposition version of ToT as ToT(Ori); we show the results below:
>
> | Model| APPS-intro. (Pass Rate %) | APPS-inter. (Pass Rate %) | APPS-comp. (Pass Rate %) | APPS-intro. (Pass@1) | APPS-inter. (Pass@1) | APPS-comp. (Pass@1) | HumanEval (Pass@1) |
> |-|-|-|-|-|-|-|-|
> | **(GPT3.5)**||||||||
> | ToT(Ori) |62.56|57.97|28.00|38|25|10|76.22|
> | ToT(In our paper)| 63.21 | 63.49 | 26.30 | 37 | 33| 11| 84.15|
> | RethinkMCTS(Ours)| **67.09**|**68.65**| **29.50** | **45** |**38**| **13**| **89.02**|
> | **(GPT4o-mini)**||||||||
> | ToT(Ori) |71.03|67.84|37.17|52|46|23|92.68|
> | ToT(In our paper)| 74.34 | 71.83 |**42.50**|55| 47|27| 93.29|
> | RethinkMCTS(Ours) | **76.60**|**74.35**|**42.50**| **59** |**49**| **28**| **94.51**|
>
> It is clear that the original version of ToT, which employs task decomposition into steps followed by searching each step individually, performs relatively poorly. In contrast, the performance improves significantly when adapted to our code reasoning-based version of ToT (as the version in our paper). This highlights the effectiveness and contribution of our work in exploring the reasoning process behind code generation.
>
> [1] Huang, D., et al. Agentcoder: Multi-agent-based code generation with iterative testing and optimisation. arXiv preprint arXiv:2312.13010.

---

> ### Author Response · Authors · 2024-11-21
> **Author Response (2/2)**
>
> > Please provide more details about the comparison setting for the number of tokens across different methods.
>
> Our approach ensures the comparison fair by keeping the number of rollouts, i.e., the number of code generated during search, the same. This is following the previous work about tree search in code generation PG-TD[2] and LATS [3]. Here we present the token usage:
>
> | Model| APPS-Intro.| APPS-Intro.|APPS-Intro.| HumanEval| HumanEval|HumanEval|
> |-|-|-|-|-|-|-|
> | **(GPT4o-mini)**|Input Token(per question) |Output Token(per question)|Cost($)|Input Token(per question)|Output Token(per question)|Cost($)|
> | ToT|24799|7156|0.008|11687|7131|0.006|
> | LATS|104634|17472|0.026|12690|7403|0.006|
> | PG-TD |27827|5378|0.007| 5959|3759|0.003|
> | LDB | 61112| 1734|0.010| 13161|480|0.002|
> | RethinkOnly(Ours)|53607|10656|0.014|24673 | 7677|0.008|
> | RethinkMCTS(Ours)|123207|17863|0.029|28479|8198|0.009|
>
> RethinkMCTS primarily incurs higher costs on input tokens than other methods. However, for LLMs, the cost associated with input tokens is significantly lower than that of output tokens. For instance, in the case of GPT-4-mini, the cost is $0.15 per million input tokens versus $0.60 per million output tokens.
>
> Our increase in token usage is primarily due to the introduction of block-level analysis, which includes the values of variables before and after each block. These values are obtained through code execution, resulting in longer textual inputs to the LLM. However, the feedback makes our model deliver significantly better results. It is essential to the success of the "rethink" mechanism, and it represents an important characteristic in the coding environment (there would be no such detailed feedback for the math reasoning problem). Therefore, incorporating such feedback is crucial. Like OpenAI's o1 model (which solves one problem with thousands of tokens in the hidden CoT and minutes to take[4]), our primary aim is not to optimize token count or computation time. Instead, the emphasis is on enabling LLMs to generate higher-quality reasoning processes and achieve superior reasoning outcomes.
>
> Our core contribution lies in focusing on the reasoning process in code generation tasks. This perspective was not explored before OpenAI's o1 model (our work was conducted concurrently with o1). We propose a "rethink" mechanism to refine the thought process of writing code. We believe this introduces a new research focus for code generation tasks: emphasizing the thought process behind writing code rather than merely aiming to search for the correct codes simply.
>
>
> > One part of the novelty is the block-level feedback, however, there is no example given for what this type of feedback looks like and no explanation about implementation details. Please provide more information.
>
> In Appendix B.5, we provide an example of verbal feedback, which includes a sample of block-level analysis. The concept of block-level analysis is inspired by prior work [5]. The basic principle involves transforming the code into a control flow graph. When a failed test case is input, it generates the execution trace corresponding to that failed test case, along with the variable values before and after each node on the trace. Each node in the trace represents a block, and by analyzing the changes in variable values before and after the execution of each block, we can assess the correctness of each block.
>
> [1] Huang, D., et al. Agentcoder: Multi-agent-based code generation with iterative testing and optimisation. arXiv preprint arXiv:2312.13010.
>
> [2] Zhang, S., et al. Planning with large language models for code generation. ICLR 2023.
>
> [3] Zhou, A., et al. Language agent tree search unifies reasoning acting and planning in language models. arXiv preprint arXiv:2310.04406.
>
> [4] https://openai.com/index/learning-to-reason-with-llms/
>
> [5] Zhong, L.,et al. Ldb: A large language model debugger via verifying runtime execution step-by-step. ACL 2024.

---

> ### Author Response · Authors · 2024-11-25
> **Official Comment by Authors**
>
> Dear Reviewer,
>
> Thank you once again for your insightful comments and valuable suggestions.
>
> We hope that our response has adequately addressed your concerns. If you have any further questions or feedback, please don't hesitate to let us know. We are committed to actively engaging with your input and continuously improving our paper.
>
> We look forward to receiving your further comments.
>
> Best regards,
>
> The Authors

---

### Official Review · Reviewer_Ruo1 · 2024-11-03

**Soundness:** 1
**Presentation:** 2
**Contribution:** 2
**Rating:** 3
**Confidence:** 4

**Summary:**

This work proposed a MCTS variant that allows node level self-refinement as a alternative way to ``expand'' the search tree with ``refined nodes''. This refinement method prompt an LLM to self-refine the thought and corresponding code, conditioned on a erroneous node. In addition to the public test cases, the authors incorporated ``block-level analysis'' and ``LLM self-evaluation'' as feedback signals to improve the evaluation of search rollouts.

**Strengths:**

- The problem, search in LLM code generation, is very relevant
- Ablation in Figure 3 shows each component plays certain role in the framework

**Weaknesses:**

- The evaluation strategy and refinement in tree search appears to be stitched for final performance but are totally irrelevant strategies. As the paper is titled as RethinkMCTS, adding additional feedback effort may make it more difficult for the audience to evaluate the role of ``rethink''.
    - for example, RethinkMCTS without VF (39 in Figure 3) can be lower than PG-TD (40 in Table 1) for APPS Intro. It is not clear what is the performance of rethink only, in comparison with the baselines, as the baselines are mainly about correction/refinements.


- My biggest concern is that, if my understanding is correct, since RethinkMCTS can have maximum 16 rollouts (meaning maximum 16 evaluations with public tests), it is not fair to compare ``pass@1'' to the base model.
    - A important baseline is: run 16 responses with a base model, and then filtered out those failed public tests, and then compute pass@1 using the responses which are correct for public tests for instance by randomly sample 1 multiple times and take the average.
    - If an additional evaluation is available, then another baseline could be best-of-N with such evaluation. For example, with the responses passed the public tests, one could for example use LLM to rank and select the BON

**Questions:**

- ToT appears very strong in the baselines, a more comprehensive comparison with ToT might be helpful to understand the contributions
    - According to my understanding the major differences are: RethinkMCTS has refine but ToT not, and the difference between evaluation strategy.
    - A comprehensive ablation should be conducted versus ToT:
        - There should be one more case: w Rethink only in Figure 3 so that the audience could understand how the Rethink mechanism improves ToT
        - While conducting the experiment with Rethink only, please include all difficulty levels (and both models as well),

- Minor points:
    - Please add an average pass rate/pass@1 for APPS (weighted by number of problems in each difficulty level)
    - PG-TD has 43.16 for APPS Comp. with 4o-mini but RethinkMCTS's 42.50 was bold instead.

---

> ### Author Response · Authors · 2024-11-21
> **Author Response (1/2)**
>
> Thank you for dedicating your time and effort to our work.
>
> > As the paper is titled as RethinkMCTS, adding additional feedback effort may make it more difficult for the audience to evaluate the role of ``rethink''.
>
> The primary contribution of our work is indeed the "rethink" mechanism. One of the motivations we propose "rethink" is effectively leveraging the execution feedback from code environment. Empirically, the experiments have confirmed the effectiveness of verbal feedback in enhancing the performance of RethinkMCTS.
>
> > RethinkMCTS can have maximum 16 rollouts (meaning maximum 16 evaluations with public tests), it is not fair to compare ``pass@1'' to the base model.
>
> We present the results for the base model rollout 16 here.
>
> |Model|APPS-Intro.(PR%)|APPS-Inter.(PR%)|APPS-Comp. (PR%)|APPS-Intro. (Pass@1)|APPS-Inter. (Pass@1)|APPS-Comp. (Pass@1)|HumanEval (Pass@1)|
> |-|-|-|-|-|-|-|-|
> | **(GPT3.5)**||||||||
> | GPT3.5 (1)| 50.43 | 40.57 | 23.67| 29| 19 | 9 | 70.12|
> | GPT3.5 (16)| 66.77 | 62.65  | 25.50| **45** | 34  | 9| 81.71|
> | RethinkMCTS (16)| **67.09** | **68.65**| **29.5**| **45** |**38**| **13**| **89.02**|
> | **(GPT4o-mini)**||||||||
> | GPT4o-mini (1)| 56.56  | 52.40 | 35  | 35  | 29 | 16 |87.20|
> | GPT4o-mini (16)| 67.79| 66.25| 38.50| 47| 41| 21 | 93.29  |
> | RethinkMCTS (16)|**76.60**| **74.35**| **42.5**| **59**| **49**| **28**| **94.51**|
>
> The pass@ 1 metric is obtained by using tree search to identify one and only one code solution, which is then tested on private test cases. The term "base model" refers to the outcome when the model independently performs simple reasoning and submits one code. To ensure a fair comparison, we followed the same evaluation approach commonly used in other tree search works [1, 2], which are usually compared directly with the base model.
>
> > About ToT baseline implementation.
>
> > A comprehensive ablation should be conducted versus ToT: Rethink only without feedback.
>
> We would like to clarify some key points about ToT:
> 1. The search for reasoning/thought process for code is one of our key contributions. The strong performance of ToT is largely because we did not implement it strictly as described in the original paper. While "thoughts" are mentioned in its title, the original paper would decompose the task into fixed steps and search for solutions within each step. In our implementation, we merged ToT with our method by defining the search space in terms of reasoning "thoughts." This significantly improved its performance, making the ToT implementation in the paper an ablation of our method. The implementation approach likely plays a significant role in determining its performance. For instance, in the AgentCoder paper[3], ToT performs poorly, whereas when integrated with our searching reasoning process, it achieves much better results.
> 2. Our model demonstrates a clear advantage, especially on models like GPT-3.5, which have weaker coding capabilities. This advantage persists even in the rethink-only setting.
>
> To better illustrate our point, we strictly follow the paper ToT and adapt a task-decomposition version as ToT(Ori).
>
> We also add RethinkOnly experiments and present the results:
>
> | Model| APPS-intro. (Pass Rate %) | APPS-inter. (Pass Rate %) | APPS-comp. (Pass Rate %) | APPS-intro. (Pass@1) | APPS-inter. (Pass@1) | APPS-comp. (Pass@1) | HumanEval (Pass@1) |
> |-|-|-|-|-|-|-|-|
> | **(GPT3.5)**||||||||
> | ToT(Ori) |62.56|57.97|28.00|38|25|10|76.22|
> | ToT(In our paper)| 63.21 | 63.49 | 26.30 | 37 | 33| 11| 84.15|
> | LATS| 54.06 | 45.86| 21.83| 36 | 20 | 7| 79.88|
> | PG-TD| 60.89| 50.80 | 26.50| 40| 25| 8| 76.22|
> | RethinkOnly(Ours)| **70.22**| **69.37**|28.33| 44| **39**| 12 | 86.59|
> | RethinkMCTS(Ours)| 67.09|68.65| **29.50** | **45** | 38| **13**| **89.02**|
> | **(GPT4o-mini)**||||||||
> | ToT(Ori) |71.03|67.84|37.17|52|46|23|92.68|
> | ToT(In our paper)| 74.34 | 71.83 | 42.50 |55| 47|27| 93.29|
> | LATS| 69.46| 67.65| 35.83 | 50 | 45| 19| 93.29|
> | PG-TD | 65.87 | 70.37| **43.16** | 45 | 46| 25| 91.46|
> | RethinkOnly(Ours) | 74.79| **75.15**| 41.67| 55  | **50**| 26| 93.90|
> | RethinkMCTS(Ours) | **76.60**|74.35| 42.50| **59** | 49 | **28**| **94.51**|
>
> The original version of ToT, which employs task decomposition into steps followed by searching each step individually, performs relatively poorly. In contrast, the performance improves significantly when adapted to our reasoning-based version of ToT (as the version in our paper). This highlights the effectiveness and contribution of our work in exploring the reasoning process behind code generation.
>
> [1] Zhang, S., et al. Planning with large language models for code generation. ICLR 2023.
>
> [2] Zhou, A., et al. Language agent tree search unifies reasoning acting and planning in language models. PMLR 2024.
>
> [3] Huang, D., et al. Agentcoder: Multi-agent-based code generation with iterative testing and optimisation. arXiv preprint arXiv:2312.13010.

---

> ### Author Response · Authors · 2024-11-21
> **Author Response (2/2)**
>
> > Minor points: Add average pass rate/pass@1 for APPS; PG-TD has 43.16 for APPS Comp. with 4o-mini but RethinkMCTS's 42.50 was bold instead.
>
> Thank you for your suggestion. Here we present the average performance of several competitive methods. And we will add these columns in the final version.
> | Model| APPS(Avg Pass Rate%) | APPS (Avg Pass@1) | HumanEval (Pass@1) |
> |-|-|-|-|
> | **(GPT3.5)**||||
> | ToT(Ori)|49.51|24.33|76.22|
> | ToT(In our paper)| 51.00 | 27.00  | 84.15|
> | LATS| 40.58 | 21.00| 79.88|
> | PG-TD|46.06|24.33| 76.22|
> | RethinkMCTS| **55.06**|**31.67**| **89.02**|
> | **(GPT4o-mini)**||||
> | ToT(Ori)|56.68|40.33|92.68|
> | ToT(In our paper)| 62.89 | 43.00|93.29|
> | LATS| 57.65|38.00| 93.29|
> | PG-TD | 59.80 | 39.33| 91.46|
> | RethinkMCTS | **64.48**| **45.33**| **94.51**|
>
> For the PG-TD's 43.16, it should be bold and we would fix it at the revision.

---

> ### Author Response · Authors · 2024-11-25
> **Official Comment by Authors**
>
> Dear Reviewer,
>
> Thank you once again for your insightful comments and valuable suggestions.
>
> We hope that our response has adequately addressed your concerns. If you have any further questions or feedback, please don't hesitate to let us know. We are committed to actively engaging with your input and continuously improving our paper.
>
> We look forward to receiving your further comments.
>
> Best regards,
>
> The Authors

---

### Author Response · Authors · 2024-11-21
**General Response**

Thank you again for all your valuable feedback and constructive suggestions. Based on the feedback from reviewers, we have identified some common concerns. Here, we provide a general response to address these shared questions and clarify our approach:

> Regarding the performance improvements of our method compared to baselines, particularly compared with ToT.

We would like clarify some key points about ToT:
1. The search for reasoning/thought process for code is one of our key contributions. The strong performance of ToT is largely because we did not implement it strictly as described in the original paper. While "thoughts" are mentioned in its title, the original paper would decompose the task into fixed steps and search for solutions within each step. In our implementation, we merged ToT with our method by defining the search space in terms of reasoning "thoughts." This significantly improved its performance, making the ToT implementation in the paper an ablation of our method. The implementation approach likely plays a significant role in determining its performance. For instance, in the AgentCoder paper[1], ToT performs poorly, whereas when integrated with our searching reasoning process, it achieves much better results.
2. Our model demonstrates a clear advantage, especially on models like GPT-3.5, which have weaker coding capabilities. This advantage persists even in the rethink-only setting.

To better illustrate our point, we strictly follow the paper of ToT and adapt a task-decomposition version of ToT as ToT(Ori), and we also add the RethinkOnly as a comparison:

| Model| APPS-intro. (Pass Rate %) | APPS-inter. (Pass Rate %) | APPS-comp. (Pass Rate %) | APPS-intro. (Pass@1) | APPS-inter. (Pass@1) | APPS-comp. (Pass@1) | HumanEval (Pass@1) |
|-|-|-|-|-|-|-|-|
| **(GPT3.5)**||||||||
| ToT(Ori) |62.56|57.97|28.00|38|25|10|76.22|
| ToT(In our paper)| 63.21 | 63.49 | 26.30 | 37 | 33| 11| 84.15|
| RethinkMCTS(Ours)| **67.09**|**68.65**| **29.50** | **45** | **38**| **13**| **89.02**|
| **(GPT4o-mini)**||||||||
| ToT(Ori) |71.03|67.84|37.17|52|46|23|92.68|
| ToT(In our paper)| 74.34 | 71.83 | **42.50** |55| 47|27| 93.29|
| RethinkMCTS(Ours) | **76.60**|**74.35**| **42.50**| **59** |**49**| **28**| **94.51**|

The original version of ToT, which employs task decomposition into steps followed by searching each step individually, performs relatively poorly. In contrast, the performance improves largely when we adapt ToT to our code reasoning-based version (as the version in our paper). This highlights the effectiveness and contribution of our work in exploring the reasoning process behind code generation.

> Regarding the time and token cost.

RethinkMCTS consumes more tokens and time due to the inclusion of verbal feedback, specifically the block-level analysis. However, the feedback makes our model deliver significantly better results. It is essential to the success of the "rethink" mechanism, and it represents an important characteristic in the coding environment (there would be no such detailed feedback for the math reasoning problem). Therefore, incorporating such feedback is crucial. Like OpenAI's o1 model (which solves one problem with thousands of tokens in the hidden CoT and minutes to take[2]), our primary aim is not to optimize token count or computation time. This "slow thinking" mode emphasizes enabling LLMs to generate higher-quality reasoning processes and achieve superior outcomes.

Our core contribution lies in focusing on the reasoning process in code generation tasks. This perspective was not explored in code generation tasks before OpenAI's o1 model (our work was conducted concurrently with o1). We propose a "rethink" mechanism to refine the thought process involved in generating code. We believe this introduces a new research focus for code generation tasks: emphasizing the thought process behind writing code rather than merely aiming to simply search for the correct codes.

We sincerely thank all the reviewers for their feedback. We hope that our responses address some of your concerns and provide clarity on the contributions of this work. We look forward to your further comments and discussions.

[1] Huang, D., et al. Agentcoder: Multi-agent-based code generation with iterative testing and optimisation. arXiv preprint arXiv:2312.13010.

[2] https://openai.com/index/learning-to-reason-with-llms/

---

### Meta-Review · Area_Chair_19HN · 2024-12-23

**Metareview:**

The reviewers agree that this is an interesting topic, but differ in how novel they consider the paper to be. Crucially, however, the reviewers have concerns about the evaluation. This boils down both to how significant the improvement is and how fair the comparison is. The improvement of the new method over the authors' own version of Tree-of-Thought is small and may not be significant (there does not appear to be any statistical significance testing done). The authors point out that the improvements over "original" ToT are larger, and that they first tested against their own, improved ToT. This does not really make things any better for the method that is supposed to be the contribution of the paper - the authors seem to just have invented an unnecessarily strong baseline. Normally, I would not hold this against them, if it wasn't for RethinkMCTS being much more resource-intensive (it uses more tokens). Some reviewers argue that this means that the comparison is not fair. I agree. If there was a convincing performance improvement, I think the additional token usage could be justified, but the lack of convincing improvement over ToT combined with much higher token usage makes the performance less impressive.

**Additional Comments On Reviewer Discussion:**

The authors engaged well with the reviewers.

---

### Decision · Program_Chairs · 2025-01-22

Reject